# Utility of the SARC-F Questionnaire for Sarcopenia Screening in Patients with Chronic Liver Disease: A Multicenter Cross-Sectional Study in Japan

**DOI:** 10.3390/jcm10153448

**Published:** 2021-08-03

**Authors:** Tatsunori Hanai, Atsushi Hiraoka, Makoto Shiraki, Ryosuke Sugimoto, Nobuhito Taniki, Akira Hiramatsu, Nobuhiro Nakamoto, Motoh Iwasa, Kazuaki Chayama, Masahito Shimizu

**Affiliations:** 1Department of Gastroenterology/Internal Medicine, Gifu University Graduate School of Medicine, Gifu 501-1194, Japan; mshiraki-gif@umin.ac.jp (M.S.); shimim@gifu-u.ac.jp (M.S.); 2Gastroenterology Center, Ehime Prefectural Central Hospital, Matsuyama 790-0024, Japan; hirage@m.ehime-u.ac.jp; 3Department of Gastroenterology and Hepatology, Mie University Graduate School of Medicine, Tsu 514-8507, Japan; sugiryo@clin.medic.mie-u.ac.jp (R.S.); motoh@clin.medic.mie-u.ac.jp (M.I.); 4Division of Gastroenterology and Hepatology, Department of Internal Medicine, Keio University School of Medicine, Tokyo 160-8582, Japan; nobuhitotaniki@keio.jp (N.T.); nobuhiro@keio.jp (N.N.); 5Department of Gastroenterology and Metabolism, Applied Life Science, Institute of Biomedical and Health Science, Hiroshima University, Hiroshima 734-8553, Japan; akirah@hiroshima-u.ac.jp; 6Collaborative Research Laboratory of Medical Innovation, Graduate School of Biomedical and Health Sciences, Hiroshima University, Hiroshima 734-8553, Japan; chayama@hiroshima-u.ac.jp

**Keywords:** muscle strength, muscle mass, SARC-F, sarcopenia, screening

## Abstract

Diagnosing sarcopenia is challenging. This multicenter cross-sectional study aimed to evaluate the utility of the SARC-F score system for identifying sarcopenia in patients with chronic liver disease (CLD). We enrolled 717 patients from five participating centers who completed the SARC-F between November 2019 and March 2021. Sarcopenia was diagnosed based on the Japan Society of Hepatology Working Group on Sarcopenia in Liver Disease Consensus. Muscle strength was estimated using a grip dynamometer, and muscle mass was assessed using computed tomography or bioelectrical impedance analysis. The association between SARC-F and sarcopenia was analyzed using a logistic regression model. The optimal SARC-F cutoff value for identifying sarcopenia was determined using receiver operating characteristic (ROC) curve analysis. Of the 676 eligible patients, 15% were diagnosed with sarcopenia. The SARC-F distribution was 0 points in 63% of patients, 1 point in 17%, 2 points in 7%, 3 points in 4%, and ≥4 points in 8%. The SARC-F items of “Strength” (odds ratio (OR), 1.98; 95% confidence interval (CI), 1.03–3.80) and “Falls” (OR, 2.44; 95% CI, 1.48–4.03) were significantly associated with sarcopenia. The SARC-F value of 1 point showed a higher discriminative ability for identifying sarcopenia than the 4 points that are conventionally used (*p* < 0.001), with an area under the ROC curve of 0.68, sensitivity of 0.65, specificity of 0.68, positive predictive value of 0.27, and negative predictive value of 0.92. SARC-F is useful for identifying patients with CLD who are at risk of sarcopenia.

## 1. Introduction

Sarcopenia is a progressive and generalized skeletal muscle disorder characterized by an age-related decline in muscle strength, muscle quantity, and/or physical performance [1,2,3]. Sarcopenia is highly prevalent in older adults and in patients with cancer or chronic disease, and it has a detrimental effect on clinical outcomes, including falls, fractures, physical disability, cognitive impairment, and mortality [1,2]. Since the prevalence of sarcopenia in patients with chronic liver disease (CLD) is likely to be higher than that in other diseases [4], the diagnosis of sarcopenia is important for establishing treatment strategies and predicting outcomes in patients with CLD.

The diagnosis of sarcopenia requires the estimation of muscle mass—a process that can be rather challenging in daily clinical practice. Methods that can specifically and precisely assess muscle mass include computed tomography, magnetic resonance imaging, dual-energy X-ray absorptiometry, and/or bioelectrical impedance analysis [1,2]. However, the widespread use of these tools is hampered by the need for expensive equipment, the time-consuming nature of these procedures, and their relative inaccessibility [1]. Sarcopenia is a potentially reversible disease that can be prevented or ameliorated by treatments such as exercise and nutritional interventions [5,6]. Simple and convenient screening tools that can be repeated regularly in routine clinical practice are therefore needed for the rapid diagnosis of sarcopenia and for the formulation of subsequent treatment decisions.

The SARC-F score system is widely used for sarcopenia screening in older adults [1]. This self-reported questionnaire is based on the cardinal features or consequences of sarcopenia [7]. The SARC-F has not only been shown to have good internal consistency and [8], but it can also predict sarcopenia-related adverse outcomes in older adults [9,10,11]. The guidelines recommend the use of the SARC-F as a first step toward identifying individuals at high risk of sarcopenia [1,2] although much remains to be elucidated about the use of the SARC-F in patients with CLD.

The aim of this multicenter cross-sectional study in Japan was to identify the optimal SARC-F cutoff value for sarcopenia screening and to clarify the relationship between SARC-F and sarcopenia.

## 2. Materials and Methods

### 2.1. Study Design

We performed a multicenter, cross-sectional study of 717 patients with CLD between November 2019 and March 2021. This study involved five institutions, including Gifu University Hospital, Ehime Prefectural Central Hospital, Mie University Hospital, Keio University Hospital, and Hiroshima University Hospital. The study objectives were explained to the participants by their healthcare providers, and the participants’ consent was implied by the return of the SARC-F questionnaire. Deidentified clinical information collected from each participating institution was used for the data analysis. The study protocol was reviewed and approved by the Institutional Review Board of the Gifu University Graduate School of Medicine (approval number: 2019-196) and the independent ethics committees of each participating institution prior to study initiation, and the study was performed in accordance with the ethical standards laid down in the Declaration of Helsinki and its later amendments.

### 2.2. Study Sample

The participants were recruited from five participating institutions. The diagnosis of cirrhosis was made primarily by the individual hepatologist at each institution and was based on the degree of liver fibrosis (estimated by a FiB-4 index > 3.25) [12], laboratory variables, clinical features of portal hypertension, medical imaging features, and/or, if available, histological features. The severity of liver disease was estimated using the Child–Pugh classification [13], and hepatocellular carcinoma (HCC) was diagnosed based on histological features or typical imaging characteristics [14]. The inclusion criteria were patients with CLD who were aged 20 years or older and who had been assessed for muscle mass and strength within 6 months of the SARC-F. Exclusion criteria included refusal to participate, pregnant women, missing data on muscle mass and/or strength measurements, and any life-threatening acute illness, including severe sepsis, heart failure, respiratory failure, and/or renal failure. Baseline characteristics within 1 month of the SARC-F were obtained from the electronic medical records of each participating site using a standardized data collection template.

### 2.3. SARC-F Questionnaire

The translated Japanese version of the SARC-F questionnaire was administered to participants according to the literature [7,15]. The SARC-F consists of five items: (1) Strength: How much difficulty do you have in lifting and carrying 4.5 kg?; (2) Assistance in walking: How much difficulty do you have walking across a room?; (3) Rising from a chair: How much difficulty do you have transferring from a chair or bed?; (4) Climbing stairs: How much difficulty do you have climbing a flight of 10 stairs?; and (5) Falls: How many times have you fallen in the past year? Each item is scored on a scale of 0 (best) to 2 (worst). Consequently, the total score ranges from 1 to 10 points, with higher values (SARC-F ≥ 4) indicating a higher risk of sarcopenia [7].

### 2.4. Sarcopenia

Patients with both low muscle mass and strength were diagnosed with sarcopenia based on the Japan Society of Hepatology Working Group on Sarcopenia in Liver Disease Consensus [3]. Muscle mass was assessed in three ways: the skeletal muscle index (SMI), the psoas muscle index (PMI), and the appendicular skeletal muscle mass index (ASMI), depending on the circumstances of each participating institution [16,17,18]. SMI was estimated using the cross-sectional area of the abdominal skeletal muscles at the third lumbar vertebra (L3) level on computed tomography. PMI was estimated in the bilateral psoas muscle area at the L3 level on computed tomography. ASMI was estimated using bioelectrical impedance analysis. There was one participating institution that used SMI to estimate low muscle mass, one used PMI, and three used ASMI. The sex-specific cutoff values for SMI that indicated that measurements for low muscle mass were considered to be 42 cm^2^/m^2^ for men and 38 cm^2^/m^2^ for women, those for PMI were 6.36 cm^2^/m^2^ for men and 3.92 cm^2^/m^2^ for women, and those for ASMI were 7.0 kg/m^2^ for men and 5.7 kg/m^2^ for women [3,19]. Grip strength was estimated using a digital Smedley dynamometer (T.K.K.5101 GRIP-D; Takei Scientific Instruments, Niigata, Japan) in a standing position with full elbow extension. The sex-specific cutoff values for grip strength that indicated low muscle strength were 26 kg for men and 18 kg for women [3].

### 2.5. Statistics

Continuous variables were presented as the median and interquartile range, and groups were compared using the Mann–Whitney *U* test. Categorical variables were shown as the number of patients and percentage (%), and groups were compared using the chi-square test. Receiver operating characteristic (ROC) curve analysis and Youden’s index were used to determine the optimal cutoff value of the SARC-F for identifying sarcopenia [20], and the results were presented as the area under the ROC curve (AUC), sensitivity, specificity, positive predictive value (PPV), and negative predictive value (NPV). AUC values were compared among different cutoff values [20]. Internal consistency reliability was assessed using Cronbach’s alpha. The relationship between the SARC-F items and sarcopenia was analyzed using logistic regression models, and the results were shown as odds ratios (ORs) with 95% confidence intervals (95% CIs). The significance threshold was set at *p* < 0.05. All analyses were performed using JMP version 9.0.2 software (SAS Institute Inc., Cary, NC, USA).

## 3. Results

### 3.1. Patients’ Characteristics

Of the original cohort of 717 participants, 41 were excluded from the study due to a lack of data on muscle mass and strength. Of the remaining 676 patients, 458 (68%) were male with a median age of 71 years and a body mass index of 23.4 kg/m^2^. In these patients, CLD was attributable to the hepatitis B virus (15%), the hepatitis C virus (31%), alcohol-related liver disease (16%), and other causes (38%). Of the enrolled patients, 425 (63%) had HCC, and 421 (62%) had liver cirrhosis. Among the patients with cirrhosis, the Child–Pugh class distribution was Child–Pugh class A (72%), B (23%), and C (5%).

Sarcopenia was diagnosed in 101 patients (15%). Patients with sarcopenia were older, more likely to be female, less obese, and had a higher prevalence of cirrhosis and HCC than those without sarcopenia (Table 1). Muscle strength and mass were significantly lower in patients with sarcopenia than in those without sarcopenia (Appendix A).

### 3.2. SARC-F and Sarcopenia

Patients with sarcopenia had worse scores on all items of the SARC-F than those without sarcopenia (all, *p* < 0.001; Table 1). Cronbach’s alpha was 0.78, indicating that the SARC-F had good internal consistency. Univariate analysis showed that each SARC-F item was significantly associated with sarcopenia (all *p* < 0.001; Table 2). Multivariate analysis showed that “Strength” (OR, 1.98; 95% CI, 1.03–3.80; *p* = 0.042) and “Falls” (OR, 2.44; 95% CI, 1.48–4.03; *p* < 0.001) were significantly associated with sarcopenia.

### 3.3. SARC-F and Clinical Characteristics

The SARC-F score distribution was 0 points in 428 (63%) patients, 1 point in 117 (17%), 2 points in 49 (7%), 3 points in 27 (4%), and ≥4 points in 55 (8%). The distribution of the SARC-F scores and the frequency of sarcopenia in each SARC-F score are shown in Figure 1.

The participants were divided into two groups based on the SARC-F cutoff value of 4 points, which indicates the possibility of sarcopenia [7]. As listed in Table 3, patients with a SARC-F ≥ 4 were older, more likely to be female, and had more advanced liver disease with respect to the prevalence of cirrhosis and the Child–Pugh class than those with a SARC-F < 4.

Patients with a SARC-F ≥ 4 also had a higher proportion of low muscle strength and sarcopenia than those with a SARC-F < 4, whereas no significant difference was noted in low muscle mass between the two groups. Muscle strength and mass (by PMI and ASMI) were also significantly lower in patients with a SARC-F ≥ 4 than in those with a SARC-F < 4 (Appendix A).

### 3.4. SARC-F for Sarcopenia Diagnosis

The SARC-F cutoff value of 4 points had an AUC of 0.57 (95% CI, 0.53–0.62) for identifying sarcopenia, and its sensitivity, specificity, PPV, and NPV were 0.21, 0.94, 0.38, and 0.87, respectively (Table 4).

ROC curve analysis showed that the SARC-F value of 1 point had the highest discriminative ability to identify sarcopenia, with an AUC of 0.68 (95% CI, 0.63–0.74; Figure 2) and sensitivity, specificity, PPV, and NPV of 0.65, 0.68, 0.27, and 0.92, respectively (Table 4). The SARC-F value of 1 point had a significantly higher AUC than that of 4 points (*p* < 0.001). Additionally, subgroup analyses on gender, age, and diagnostic methods for sarcopenia showed that the use of SARC-F was comparable in each subgroup (Appendix A).

## 4. Discussion

Sarcopenia is associated with adverse clinical outcomes in patients with CLD, but it can be reversed with individualized interventions such as exercise and nutritional therapy [13,21,22,23]. Therefore, appropriate screening and early detection of sarcopenia may prevent adverse outcomes and provide timely treatment to patients with sarcopenia. Several methods have been proposed for the diagnosis of sarcopenia, but their use in routine clinical practice is limited by cost and the time-consuming nature of the procedures. We previously showed the usefulness of the finger-circle test as screening for muscle atrophy in CLD patients [24]. The predictive ability of this test (AUC of 0.65, sensitivity of 0.62, specificity of 0.67) is similar to those of the present study. Other studies have shown that the arm and calf circumferences have a better ability to predict sarcopenia (AUC of 0.84–0.91, sensitivity of 66–88%, and specificity of 74–90%) [25]. However, this study is limited by the fact that it was a single-center study and only included CLD patients without ascites or leg edema. Although the SARC-F is widely used and recommended for sarcopenia screening in community-dwelling older adults and in some diseases [1], whether these results can be translated into patients with CLD has not yet been elucidated. This multicenter, cross-sectional study provides substantial evidence that SARC-F has a high discriminative ability for the diagnosis of sarcopenia in patients with CLD.

SARC-F has been reported to show low sensitivity (14–21%) and high specificity (90–94%) for identifying sarcopenia in community-dwelling older adults [26]. To our knowledge, three studies involving patients with CLD have also shown that the SARC-F has low sensitivity (15–45%) and high specificity (91–98%) in this population [17,27,28]. However, the findings of these studies were limited by the small sample size, single center-study design, and the fact that the diagnosis of sarcopenia was not based on the criteria for patients with CLD. The findings of this large, multicenter study confirm previous findings and provide evidence to support the usefulness of the SARC-F in identifying patients with CLD who are at high risk for sarcopenia.

There are many possible reasons for the low sensitivity and high specificity of the SARC-F. First, since the components of the SARC-F seem to focus on the symptoms of functional decline associated with advanced sarcopenia, this screening tool may be limited in its ability to identify patients in the early stages of sarcopenia [7,29]. Second, since each item of the SARC-F focuses particularly on muscle strength rather than muscle mass, it is possible that this screening tool specifically assesses loss of muscle strength and motor function rather than loss of muscle mass [7]. It is generally recognized that loss of muscle strength is a more important feature of sarcopenia than the loss of muscle mass and is a more sensitive predictor of adverse outcomes [1,30]. Our findings also show that the SARC-F item “Strength” independently predicts sarcopenia, and that a SARC-F ≥ 4 is clearly associated with muscle strength rather than muscle mass. These results are consistent with those of previous studies showing that the SARC-F score has a higher correlation with muscle strength but a lower correlation with muscle mass [28,31,32]. Muscle strength has many advantages over muscle mass assessed by expensive machines in CLD patients. Recent evidence has shown that muscle strength is useful in predicting liver-related complications [33,34,35]. Furthermore, our previous report found that low muscle strength rather than low muscle mass was associated with an increased risk of mortality in patients with liver cirrhosis [36]. On the other hand, the advantage of the SARC-F is that it is a self-report questionnaire that is easy to complete before an outpatient visit and can be administered to many patients at risk for sarcopenia. Since the recommended SARC-F cutoff of 4 points may only identify patients with more advanced sarcopenia, a new SARC-F cutoff is needed for the early detection of sarcopenia, especially in patients with CLD.

The criteria for an ideal screening test include not only accurate sensitivity to identify patients at an increased risk of having a disease but also high specificity to minimize false-positive rates to avoid unnecessary testing [20,37]. The present study showed that the SARC-F cutoff value of 1 point provides higher sensitivity and NPV than that of 4 points that are used conventionally although the new cutoff value had a lower PPV because of the low prevalence of sarcopenia in this cohort [37]. These findings may provide meaningful implications for the use of the SARC-F in daily clinical practice. First, the increased sensitivity suggests that the new SARC-F cutoff value has an improved ability to correctly identify patients at risk of sarcopenia. Second, the high NPV suggests that patients with a SARC-F < 1 are unlikely to suffer from sarcopenia and that the further assessment of muscle mass and strength is unnecessary. Third, the low PPV suggests that patients with a SARC-F ≥ 1 may still have sarcopenia, and further evaluation is necessary for the definitive diagnosis of sarcopenia. Therefore, the SARC-F, which can be assessed repeatedly throughout the clinical course, might be a more practical and efficient way to screen for sarcopenia considering the cost and time involved. Although the results of this study showed that the SARC-F value of 1 point had a significantly higher AUC than that of the 4 points that are used conventionally, the relatively low sensitivity, specificity, and AUC are insufficient to conclude that SRAC-F is a useful method for sarcopenia screening. Further studies are needed to establish the role of SARC-F in screening for sarcopenia in patients with chronic liver disease.

This study has several limitations. First, the cross-sectional nature of this study does not allow us to prove causality between the SARC-F and sarcopenia, and the findings may be influenced by unmeasured confounders. Second, the subjective nature of each SARC-F items makes it difficult to fully accept the reliability of self-reported information. Third, differences in the methods of estimating muscle mass in the five participating centers may affect the results, limiting our conclusions about the usefulness of the SARC-F. Fourth, the results showed that the SARC-F value of 1 point had a higher sensitivity than the 4 points that are used conventionally. However, the sensitivity of the SARC-F is still low, which may be a limitation in screening for sarcopenia. Finally, sarcopenia was diagnosed based on the diagnostic criteria for sarcopenia in liver disease that is used in Japan [3]. Additionally, because our results were obtained in a real-world cohort of CLD patients treated at tertiary referral centers in Japan, the subjects in this study may differ from CLD patients in other settings in terms of age, clinical characteristics, and comorbidities. Thus, the results of this study may not be generalizable to other populations or regions. Although our findings may have meaningful clinical implications, further research is needed to determine whether the use of the SARC-F allows for more efficient screening for sarcopenia and whether it subsequently provides timely treatment to patients who are at high risk of sarcopenia.

## 5. Conclusions

In conclusion, this nationwide cohort of patients with CLD corroborates previous findings and provides evidence that the SARC-F assists in the improvement of diagnostic strategies and the early identification of patients with sarcopenia who would be the most eligible for interventions.

## Figures and Tables

**Figure 1 jcm-10-03448-f001:**
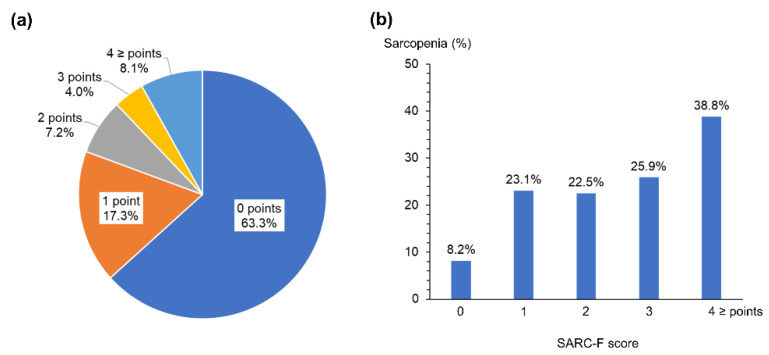
SARC-F and sarcopenia. (**a**) The distribution of SARC-F scores and (**b**) the frequency of sarcopenia in each SARC-F score in patients with chronic liver disease.

**Figure 2 jcm-10-03448-f002:**
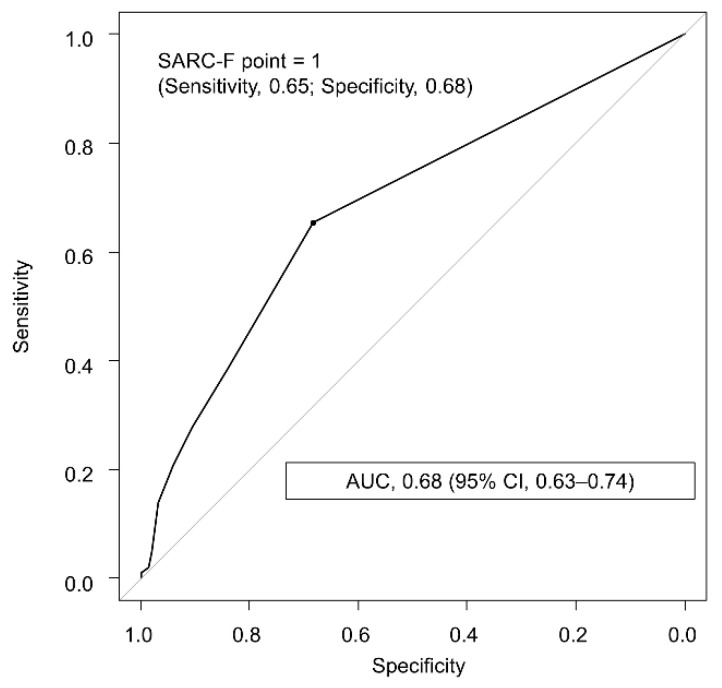
Receiver operating characteristic curve of the SARC-F for identifying sarcopenia in patients with chronic liver disease. The optimal cutoff value was determined using the highest Youden index value. AUC, area under the receiver operating characteristic curve.

**Table 1 jcm-10-03448-t001:** Baseline characteristics of patients with and without sarcopenia.

	Total Cohort	No Sarcopenia	Sarcopenia	*p*-Value *
	(*n* = 676)	(*n* = 575)	(*n* = 101)	
Characteristics				
Age, years	71 (64–78)	70 (61–77)	78 (73–84)	<0.001
Body mass index, kg/m²	23.4 (21.3–26.0)	23.7 (21.7–26.4)	21.7 (19.7–24.0)	<0.001
Male sex	458 (68)	399 (69)	59 (58)	0.037
Etiology				0.032
Hepatitis B virus	99 (15)	92 (16)	7 (7)	
Hepatitis C virus	212 (31)	171 (30)	41 (41)	
Alcohol-related	110 (16)	96 (17)	14 (14)	
Others	255 (38)	216 (38)	39 (39)	
Liver cirrhosis	421 (62)	347 (60)	74 (73)	0.014
Child–Pugh class †				0.800
A	302 (72)	251 (72)	51 (69)	
B	97 (23)	78 (23)	19 (26)	
C	22 (5)	18 (5)	4 (5)	
Hepatocellular carcinoma	425 (63)	352 (61)	73 (72)	0.034
SARC-F components				
Strength				<0.001
None (0)	577 (85)	508 (88)	69 (68)	
Some (1)	77 (11)	49 (9)	28 (28)	
A lot or unable (2)	22 (3)	18 (3)	4 (4)	
Assistance in walking				<0.001
None (0)	611 (90)	534 (93)	77 (76)	
Some (1)	58 (9)	36 (6)	22 (22)	
A lot, use aids, or unable (2)	7 (1)	5 (1)	2 (2)	
Rising from a chair				<0.001
None (0)	595 (88)	522 (91)	73 (72)	
Some (1)	78 (12)	51 (9)	27 (27)	
A lot or unable without help (2)	3 (0)	2 (0)	1 (1)	
Climbing stairs				<0.001
None (0)	529 (78)	468 (81)	61 (60)	
Some (1)	126 (19)	93 (16)	33 (33)	
A lot or unable (2)	21 (3)	14 (2)	7 (7)	
Falls				<0.001
None (0)	551 (82)	487 (85)	64 (63)	
1–3 falls (1)	120 (18)	84 (15)	36 (36)	
4 or more falls (2)	5 (1)	4 (1)	1 (1)	

Values are presented as number (percentage) or median (interquartile range). † Patients with cirrhosis were evaluated. * The chi-square test for categorical variables or Mann–Whitney U test for continuous variables were used to compare clinical characteristics between the two groups.

**Table 2 jcm-10-03448-t002:** The relationship between the SARC-F items and sarcopenia.

	Univariate *		Multivariate *	
Characteristics	OR (95% CI)	*p*-Value	OR (95% CI)	*p*-Value
Strength †	3.52 (2.15–5.74)	<0.001	1.98 (1.03–3.80)	0.042
Assistance in walking †	4.06 (2.32–7.09)	<0.001	1.47 (0.62–3.48)	0.376
Rising from a chair †	3.78 (2.25–6.35)	<0.001	1.56 (0.73–3.34)	0.249
Climbing stairs †	2.87 (1.83–4.50)	<0.001	1.10 (0.56–2.16)	0.771
Falls ‡	3.20 (2.01–5.09)	<0.001	2.44 (1.48–4.03)	<0.001

* Logistic regression analysis. † Some/unable vs. none. ‡ Falls vs. none. CI, confidence interval; OR, odds ratio.

**Table 3 jcm-10-03448-t003:** Comparison of baseline characteristics of patients with a SARC-F < 4 and those with ≥4.

	SARC-F < 4	SARC-F ≥ 4	*p*-Value *
Characteristics	(*n* = 621)	(*n* = 55)	
Age, years	71 (62–78)	79 (70–83)	<0.001
Body mass index, kg/m²	23.4 (21.3–26.1)	23.6 (21.7–25.9)	0.923
Male sex	433 (70)	25 (46)	<0.001
Etiology			0.577
Hepatitis B virus	92 (15)	7 (13)	
Hepatitis C virus	195 (31)	17 (31)	
Alcohol-related	104 (17)	6 (11)	
Others	230 (37)	25 (46)	
Liver cirrhosis	375 (60)	46 (84)	<0.001
Child-Pugh class †			0.013
A	277 (74)	25 (54)	
B	81 (22)	16 (35)	
C	17 (5)	5 (11)	
Hepatocellular carcinoma	388 (63)	37 (67)	0.561
Body composition measurements			
Low muscle strength	168 (27)	43 (78)	<0.001
Low muscle mass	308 (50)	27 (49)	1.000
Sarcopenia	80 (13)	21 (38)	<0.001

Values are presented as number (percentage) or median (interquartile range). † Patients with cirrhosis were evaluated. * The chi-square test for categorical variables or Mann–Whitney U test for continuous variables were used to compare clinical characteristics between the two groups.

**Table 4 jcm-10-03448-t004:** Sensitivity, specificity, and positive and negative predictive values for sarcopenia.

	Sensitivity (95% CI)	Specificity (95% CI)	PPV (95% CI)	NPV (95% CI)
SARC-F ≥ 1	0.65 (0.55–0.75)	0.68 (0.64–0.72)	0.27 (0.21–0.33)	0.92 (0.89–0.94)
SARC-F ≥ 2	0.39 (0.29–0.49)	0.84 (0.81–0.87)	0.30 (0.22–0.38)	0.89 (0.86–0.91)
SARC-F ≥ 3	0.28 (0.19–0.38)	0.91 (0.88–0.93)	0.34 (0.24–0.45)	0.88 (0.85–0.90)
SARC-F ≥ 4	0.21 (0.13–0.30)	0.94 (0.92–0.96)	0.38 (0.25–0.52)	0.87 (0.84–0.90)

CI, confidence interval; NPV, negative predictive value; PPV, positive predictive value.

## Data Availability

The data presented in this study are available upon request from the corresponding author.

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
