# Peer review of "Utility of the SARC-F Questionnaire for Sarcopenia Screening in Patients with Chronic Liver Disease: A Multicenter Cross-Sectional Study in Japan"

_jcm, 2021, doi:10.3390/jcm10153448_

Round 1

Reviewer 1 Report

Comments to the Author 
In this study, the authors demonstrated the usefulness of the SARC-F for screening sarcopenia in CLD patients.

Major comments:
1.    Since expensive machines such as CT and MRI were needed to diagnose sarcopenia, the authors argue that a screening method was needed. Since the SARC-F was related to muscle strength rather than muscle mass in CLD patients, why not just assess muscle strength without using the SARC-F? I don't think an expensive machine is necessary if only muscle strength is assessed.
2.    Was the population of this study consistent with the general population of CLD patients?

Author Response

Responses to Reviewer 1

Thank you very much for reviewing our manuscript and offering valuable advice. We appreciate your comments, which have helped us to improve our manuscript. Please find below detailed responses to the reviewer’s comments.

Major comments:

  1. Since expensive machines such as CT and MRI were needed to diagnose sarcopenia, the authors argue that a screening method was needed. Since the SARC-F was related to muscle strength rather than muscle mass in CLD patients, why not just assess muscle strength without using the SARC-F? I don't think an expensive machine is necessary if only muscle strength is assessed.

We really agree that muscle strength has many advantages over muscle mass assessed by expensive machines in CLD patients. Recent evidence has shown that muscle strength is useful in predicting liver-related complications (1-3). Furthermore, our previous report found that low muscle strength rather than low muscle mass was associated with an increased risk of mortality in patients with liver cirrhosis (4). On the other hand, the advantage of the SARC-F is that it is a self-report questionnaire that is easy to complete before an outpatient visit and can be administered to many patients at risk of sarcopenia. We have added this information along with the relevant references in the revised manuscript (Page 8, lines 241–248; and new References #33–#36). We hope that this explanation and the addition of references meet with the approval of the reviewer. Thank you for the valuable suggestion.

  1. Huisman EJ, Trip EJ, Siersema PD, van Hoek B, van Erpecum KJ. Protein energy malnutrition predicts complications in liver cirrhosis. Eur J Gastroenterol Hepatol. 2011;23(11):982-9.
  2. Merli M, Giusto M, Lucidi C, Giannelli V, Pentassuglio I, Di Gregorio V, et al. Muscle depletion increases the risk of overt and minimal hepatic encephalopathy: results of a prospective study. Metabolic brain disease. 2013;28(2):281-4.
  3. Augusti L, Franzoni LC, Santos LA, Lima TB, Ietsugu MV, Koga KH, et al. Lower values of handgrip strength and adductor pollicis muscle thickness are associated with hepatic encephalopathy manifestations in cirrhotic patients. Metabolic brain disease. 2016;31(4):909-15.
  4. Hanai T, Shiraki M, Imai K, Suetsugu A, Takai K, Moriwaki H, et al. Reduced handgrip strength is predictive of poor survival among patients with liver cirrhosis: A sex-stratified analysis. Hepatology research : the official journal of the Japan Society of Hepatology. 2019;49(12):1414-26.

  1. Was the population of this study consistent with the general population of CLD patients?

Thank you for the useful comment. Because our results were obtained in a real-world cohort of CLD patients treated at tertiary referral centers in Japan, the subjects in this study may differ from CLD patients in other settings in terms of age, clinical characteristics, and comorbidities. We have added this information in the revised manuscript (Page 8, lines 279–282). We appreciate your useful comment.

In closing, let me thank you once again for your comments which have helped us to improve the quality of our paper. We hope that the above responses meet with the approval of the editors and reviewers.

Reviewer 2 Report

Utility of the SARC-F questionnaire for sarcopenia screening in patients with chronic liver disease: a multicenter cross-sectional study in Japan

Jcm-1291838

In this study, authors evaluated the utility of the SARC-F score system for identifying sarcopenia in patients with chronic liver disease (CLD) in Japan. They concluded that SARC-F is useful for identifying patients with CLD at risk of sarcopenia.

The SARC-F is a typical screening tool for sarcopenia that is recommended in the european working group on sarcopenia in older people (EWGSOP) guidelines, and it is important to evaluate its effectiveness in Japanese.

I have some concerns.

Major points:

1 The authors report the distribution of SARC-F scores in the text, please describe the distribution of each score in a figure. In addition, please describe the frequency of sarcopenia in each SARC-F score.

2 There are several screening methods for sarcopenia that have already been reported (1, 2). For example, reports using lower leg circumference and upper arm circumference have shown higher sensitivity and specificity than the present study. Please discuss the comparison with them and highlight the advantages of SARC-F.

I think the advantage of the SARC-F is that it is a self-report questionnaire, so it is easy to complete before the outpatient visit..

(1) Hiraoka A, Nagamatsu K, Izumoto H, et al. Easy surveillance of muscle volume decline in chronic liver disease patients using finger-circle (yubi-wakka) test. J Cachexia Sarcopenia Muscle. 2019;10:347-354.

 (2) Endo K, Sato T, Kakisaka K, et al. Calf and arm circumference as simple markers for screening sarcopenia in patients with chronic liver disease. Hepatol Res. 2021;51:176-189.

Author Response

Responses to Reviewer 2

Thank you very much for reviewing our manuscript and offering valuable advice. We appreciate your comments, which have helped us to improve our manuscript. Please find below detailed responses to the reviewer’s comments.

Major points:

  1. The authors report the distribution of SARC-F scores in the text, please describe the distribution of each score in a figure. In addition, please describe the frequency of sarcopenia in each SARC-F score.

Based on the reviewer’s comment, we have provided the information on the distribution of SARC-F scores and the frequency of sarcopenia in each SARC-F score in the form of a figure (new Figure 1). Thank you for the useful suggestion.

  1. There are several screening methods for sarcopenia that have already been reported (1, 2). For example, reports using lower leg circumference and upper arm circumference have shown higher sensitivity and specificity than the present study. Please discuss the comparison with them and highlight the advantages of SARC-F.

I think the advantage of the SARC-F is that it is a self-report questionnaire, so it is easy to complete before the outpatient visit.

(1) Hiraoka A, Nagamatsu K, Izumoto H, et al. Easy surveillance of muscle volume decline in chronic liver disease patients using finger-circle (yubi-wakka) test. J Cachexia Sarcopenia Muscle. 2019;10:347-354.

 (2) Endo K, Sato T, Kakisaka K, et al. Calf and arm circumference as simple markers for screening sarcopenia in patients with chronic liver disease. Hepatol Res. 2021;51:176-189.

Thank you for the useful suggestion. Several useful methods have been proposed for sarcopenia screening. We previously showed the usefulness of the finger-circle test as screening for muscle atrophy in CLD patients (1). The predictive ability of this test (AUC of 0.65, sensitivity of 0.62, specificity of 0.67) is similar to those of the present study. Other study has shown that the arm and calf circumferences have a better ability to predict sarcopenia (AUC of 0.84–0.91, sensitivity of 66–88%, and specificity of 74–90%) (2). However, this study is limited by the fact that it was a single-center study and only included CLD patients without ascites or leg edema. As you pointed out, the advantage of the SARC-F is that it is a self-report questionnaire that is easy to complete before an outpatient visit and can be administered to many patients at risk of sarcopenia. We have added this information along with the relevant references in the revised manuscript (Page 7, lines 208–215; Page 8, lines 246–248; and new References #24 and #25). We appreciate your useful comment.

In closing, let me thank you once again for your comments which have helped us to improve the quality of our paper. We hope that the above responses meet with the approval of the editors and reviewers.

Round 2

Reviewer 2 Report

The points I pointed out have been appropriately corrected.

This manuscript is a resubmission of an earlier submission. The following is a list of the peer review reports and author responses from that submission.

Round 1

Reviewer 1 Report

I read the manuscript "Utility of the SARC-F questionnaire for sarcopenia screening in patients with chronic liver disease: a multicenter cross-sectional study in Japan” with great interest, given growing importance of sarcopenia with global population aging. However, I noted substantial drawbacks of the study limiting the importance of the study. Also, data is somehow scarce and analyses performed in the study seem less robust. Most importantly, the results could not support the role of the SARC-F questionnaire for sarcopenia screening.

  1. The authors suggested the SARC-F value of 1 point as a cutoff for identifying sarcopenia in patients with chronic liver disease, instead of 4 points used conventionally. Considering the distribution of SARC-F (from 0 to 10), there is doubt about whether the cutoff value of 1 point is appropriate.

  2. Specifically, the sensitivity and specificity of SARC-F value of 1 point were only 0.65 and 0.68, respectively, and AUC for identifying sarcopenia was 0.68 and thus not strong enough to conclude that SRAC-F is a useful method for sarcopenia screening.

  1. It is well known that sex hormones significantly affect muscle metabolism. Therefore, the role of SARC-F for sarcopenia screening should be separately analyzed in each gender.

  1. Sarcopenia is more common among people over 65 years of age. Therefore, it would be interesting to perform the subgroup analyses in participants over 65 and under.

  1. The critical issue about defining the sarcopenia is that each institution used different methods of estimating muscle mass. Therefore, the authors need to show the results according to the institutions that used different methods.

Reviewer 2 Report

Screening tests for sarcopenia are useful concepts provide it results in actionable information. The authors demonstrate the sensitivity/specificity of the SARC-F score for sarcopenia using specific L3 level psoas cutoffs. This test is of low sensitivity and high spec in others' hands and that is what is found here. The novelty is the size of the cohort.

  1. A low sensitivity test is of limited value in this setting. The scan is high specificity. Missing at risk patients is the problem. Perhaps the authors would like to demonstrate how to improve sensitivity, ie. by including Child class?
  2. Many of these patients may not have cirrhosis. Most are Child A and one of the criteria was FIB4>3.25 which is simply inadequate. Please provide data on how entry criteria were defined for the cohort (proportion per cirrhosis criterion)
  3. The sacopenia rate is low for cirrhosis cohorts. Isnt the SARC-F score potentially MORE meaningful than sarcopenia itself. After all if a patient tells me they are weak, that is a better phenotype than the radiographical classification. Note, no outcomes were assessed.
  4. If the authors wish to intervene on sarcopenia, please tell us what they propose and how many of these patients would be candidates for their proposed interventions. 
  5. Compare and contrast: PMID: 3129278